# GRAPH SET TRANSFORMER: LEARNING GRAPH REPRESENTATIONS WITH SET CONTEXT

**Jose Eduardo Escrig Molina** [*]
Bioprocesss Engineering Group
Wageningen University
Wageningen, 6708 PB, NL
`joseeduardo.escrigmolina@wur.nl`

**Daniel Probst**
Bioinformatics Group
Wageningen University
Wageningen, 6708 PB, NL
`daniel.probst`@wur.nl

## ABSTRACT

We introduce the Graph Set Transformer (GST), a neural network architecture for learning on sets of graphs, motivated by the assumption that structurally heterogeneous graphs may share higher-level semantics. Existing architectures, including DeepSets and Set Transformer, require pre-encoded graph embeddings from a separate GNN, creating a bottleneck between feature extraction and set-level contextualisation. In contrast, GST avoids this bottleneck by performing node-level feature propagation and cross-graph contextual modelling simultaneously, fusing the two levels of information through a gating mechanism. Across five molecular classification benchmarks, GST achieves consistent ROC-AUC scores of 98.5-99.6% compared to the best baselines of 89.3-98.8% for large sets of cardinalities 10 and 20. In a drug-drug interaction benchmark on sparse, undersampled data, GST improves the F1 score by 17.5% compared to the best baseline.

## 1 INTRODUCTION

A set is a fundamental mathematical structure and data type that provides a permutation-invariant representation of a collection of unordered elements, lacking an inherent structure such as that of sequences or vectors. This property, while making sets suitable for modelling data without or with unknown sequential structure, has also proven challenging for machine learning Vinyals et al. (2015). However, recent advances, including DeepSets, Set Transformer, or PointNet, have introduced permutation-invariant deep learning architectures suitable for sets Zaheer et al. (2017); Charles et al. (2017); Lee et al. (2019). While DeepSets and Set Transformer require data to be represented as sets of feature vectors, PointNet specialises in point clouds. Recently, Chinello & Boracchi (2025) have introduced the convolutional set transformer (CST), which learns directly on sets of images.

Here, we present a generalisation of their architecture to graphs, the graph set transformer (GST). Unlike DeepSets or Set Transformer, which require an vector-embeddings of graphs as input, GST learns directly on graphs by injecting set-level information back into node features in every layer of a graph neural network (GNN) that accepts sets of graphs as input (Figure 1). We propose that combining feature extraction and set contextualisation in this way removes a bottleneck between learning graph embeddings and learning set context, thereby improving generalisation ability and ultimately predictive performance. Code associated with this study can be found here: `https://anonymous.4open.science/r/gst-conference-33B7`.

## 2 RELATED WORK

### 2.1 DEEPSETS

DeepSets, introduced by Zaheer et al. (2017), defines a deep network architecture for learning permutation-invariant functions over sets. It is based on the theorem stating that a function $f(X)$

---

[*]Use footnote for providing further information about author (webpage, alternative address)—*not* for acknowledging funding agencies. Funding acknowledgements go at the end of the paper.

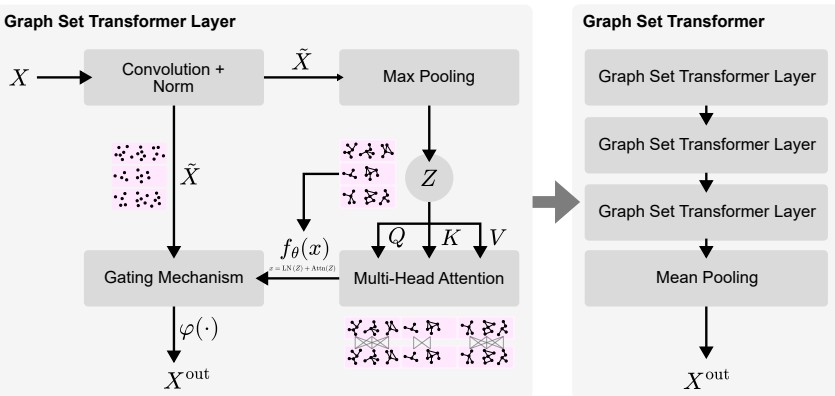

Figure 1: Graph set transformer architecture.

operating on a set $X$ is a valid set function, and invariant to the permutation of instances in $X$, iff it can be decomposed in the form $\rho\left(\sum_{x \in X} \phi(x)\right)$, where the summation can be replaced with any pooling function. Functions $\phi$ and $\rho$ are parametrised by neural networks. While the pooling-based aggregation is computationally efficient, each element is encoded independently by $\rho$. This limits the representational power of the architecture as potentially relevant information about inter-element relationships is discarded. This limitation is addressed by the Set Transformer.

## 2.2 SET TRANSFORMER

The Set Transformer architecture, introduced by Lee et al. (2019), addresses the central limitation of DeepSets by implementing an encoder-decoder structure that leverages attention mechanisms to capture inter-element relationships. The encoder consists of induced set attention blocks (ISABs), which apply a sparse Gaussian processes-inspired inducing point method to efficiently compute attention. In the decoder, Lee et al. first aggregate the latents using a pooling by multihead attention (PMA) component, followed by self-attention blocks (SABs), which enable elements in a set to attend to one another.

## 2.3 CONVOLUTIONAL SET TRANSFORMER

The convolutional set transformer (CST) was recently introduced by Chinello & Boracchi (2025) and laid the groundwork for the graph set transformer presented in this article. CST takes visu-ally heterogeneous image sets that share high-level semantics as inputs, a concept that we gener-alise from Euclidean space to non-Euclidean domains. Conceptually, this requires a shift from the pixel–image–set hierarchy to a node–neighbourhood–graph–set hierarchy, which is introduced by the graph convolutional network. We replace the additive dynamic bias, which uniformly adds a shift across spatial locations, with a more expressive learned gating mechanism that enables node-specific blending of global and local information.

## 2.4 APPLICATIONS

While the use of graphs is ubiquitous in chemistry, the use of sets has received relatively little attention. Examples include recent work on molecular set representation learning by Boulougouri et al. (2024), who used sets of atoms or molecules to predict molecular properties, reaction yields, and binding affinities, or Goldman et al. (2023), who used the Set Transformer architecture to learn mass spectra as sets of peaks. Another notable application concerns predicting chemical properties from ensembles, or sets, of molecular conformers (Zhu et al., 2024). However, within the scope of this study, we chose to avoid the introduction of geometry-aware convolutional layers. By excluding methods that account for molecular geometry, we aim to isolate the representational capacity of set-based architectures and restrict ourselves to the graph convolutional operator introduced by Kipf & Welling (2016).

## 3 METHODOLOGY

### 3.1 PROPOSED ARCHITECTURE

Let $\mathcal{S} = \{\mathcal{G}_1, \ldots, \mathcal{G}_M\}$ be a collection of sets of arbitrary graphs $\mathcal{G}_S = \{G_{s,1}, \ldots, G_{s,|\mathcal{G}_S|}\}$. Each graph $G_{s,m} = (V_{s,m}, E_{s,m})$ has a node feature matrix $X_{s,m}$ and an adjacency matrix $A_{s,m}$. Each of our graph set convolutional layers initially applies a graph convolution

$$\tilde{X}_{s,m} = \sigma \left( \hat{D}_{s,m}^{-1/2} \hat{A}_{s,m} \hat{D}_{s,m}^{-1/2} X_{s,m} W \right)$$

where $\hat{A}_{s,m} = A_{s,m} + I$ is the adjacency matrix with self-loops, $\hat{D}_{s,m}^{-1/2}$ the degree matrix of $\hat{A}_{s,m}$, $W$ the learnable weight matrix, and $\tilde{X}_{s,m} \in \mathbb{R}^{|V_{s,m}| \times d}$ are the updated node embeddings in embedding dimension $d$. We chose the convolutional layer introduced by Kipf & Welling (2016), however, it can be replaced by any implementation. The graph convolution is followed by global graph pooling

$$z_{s,m} = \text{Pool}(\{\tilde{x}_{s,m,v} | v \in V_{s,m}\})$$

where $\tilde{x}_{s,m,v} \in \mathbb{R}^d$ is the embedding vector of node $v$ and $z_{s,m} \in \mathbb{R}^d$ is the pooled graph embedding vector. We use global max pooling as the pooling function $\text{Pool}(\cdot)$ in all our experiments. For each set $\mathcal{G}_S$, the graph embedding vectors are stacked and normalised row-wise.

$$\bar{Z}_S = \text{LN} \left( \begin{bmatrix} z_{s,1} \\ \vdots \\ z_{|\mathcal{G}_S|} \end{bmatrix} \right) \in \mathbb{R}^{|\mathcal{G}_S| \times d}$$

Unlike Set Transformer, the proposed Graph Set Transformer introduces multi-head attention acting on $\bar{Z}_S$ in each graph set convolutional layer.

$$\text{Attn}(\bar{Z}_S) = \text{softmax} \left( \frac{Q_S K_S^\top}{\sqrt{d_h}} \right) V_S$$

We introduce a residual connection He et al. (2016) and a further normalisation step Ba et al. (2016). These help stabilize training and facilitate gradient flow through the network.

$$Z_S^{(1)} = \text{LN} \left( \bar{Z}_s \right) + \text{Attn}(\bar{Z}_S)$$

Following the standard transformer design Vaswani et al. (2017), we followed with a feed-forward network and an additional residual connection, mixing features across channels and increasing the model capacity.

$$Z_S^{(2)} = \text{LN} \left( Z_S^{(1)} + \text{FFN}(Z_S^{(1)}) \right)$$

Finally, the global set-level information is broadcast back to each node $v \in V_{(s,m)}$ and fused with local information through gating

$$g_{s,m,v} = \varphi \left( W_{\text{gate}} \begin{bmatrix} \bar{x}_{s,m,v} \\ s_{s,m,v} \end{bmatrix} \right),$$

$$x_{s,m,v}^{\text{out}} = s_{s,m,v} \odot s_{s,m,v} + (1 - g_{s,m,v}) \odot \bar{x}_{s,m,v}$$

where $s_{s,m,v} = Z_S^{(2)}[m]$ denotes the set-informed context vector assigned to node $v$ extracted from the $m$-th row of $Z_S^{(2)}$, $\bar{x}_{s,m,v}$ is the embedding vector of node $v$, and $x_{s,m,v}^{out} \in \mathbb{R}^d$ is the final node embedding returned by the graph set convolution layer. The rationale behind the gating mechanism is the heterogeneity of graphs compared to the regular grid structure of images, requiring node-specific blending of set context.

### 3.1.1 EXPERIMENTAL DESIGN

For all experiments, we compare GST with three layers (using `GCNConv`) as a basis to DeepSets and SetTransformer, both applied to graph embeddings produced by a graph neural network with three graph convolutional layers (`GCNConv`). The hyperparameters of all models were tuned on the BACE data set with a set size of 5. The number of hidden dimensions of all graph convolutional

layers in all models was left at 64. All experiments are set-to-global tasks, where we train on set-level labels and predict set-level labels.

Initially, we evaluate the performance of GST on the CIFAR-10 ($n = 60,000$) data set (Krizhevsky et al., 2009). We chose an image multi-class classification task to assess whether the image-based performance characteristics measured by Chinello & Boracchi (2025) compare to those of our graph-based approach. However, we do not use raw image data but graph-encoded images instead and therefore refrain from direct comparisons of accuracy magnitude (Dwivedi et al., 2023). On the CIFAR-10 data set, the models were evaluated on five different set cardinalities (1, 5, and 10). In addition, we repeat the experiments of Chinello & Boracchi (2025) and train on sets with cardinalities 5, 10, and 5-20 and predict labels for singlets during inference. All results are averaged over 5 runs with different random seeds, and we report the mean and standard deviation. A set size of 1 corresponds to processing individual samples without set aggregation, serving as a baseline. We report the macro-averaged ROC-AUC, computed using a one-vs-rest strategy.

Next, we consider molecular graphs. A central challenge in evaluating our approach is the lack of molecular benchmark data sets that group molecules into meaningful sets. Although conformer ensemble sets exist, as described by Zhu et al. (2024), we aim to initially evaluate GST on graphs without spatial features. In addition, we focus on set-to-global tasks, in which the aim is to predict a label for a whole set. Therefore, we initially group molecules from classification benchmark data sets into random sets of varying cardinality (5, 10, 20, and random cardinalities between 1-5 and 1-10). We report means across five repeated runs, keeping the sets in the validation and test sets fixed while shuffling the set composition in the training set with each run. For binary classification tasks on the data sets BACE ($n = 1,513$), BBBP ($n = 2,039$), Pgp Broccatelli ($n = 1,212$), and CYP3A4 (Substrate, $n = 667$), we report the Area Under the Receiver Operating Characteristic Curve (ROC-AUC) (Wu et al., 2018; Huang et al., 2021). Due to the small size of CYP3A4, it was only evaluated for set cardinalities of 5 and 1-5.

Finally, we evaluate our method on a multi-class drug-drug (n=191,808) interaction benchmark data set (Huang et al., 2021). We consider the pairs of drugs in the data set as sets of cardinality 2. For this multi-class task, we report accuracy, F1 (macro), and area under the precision-recall curve (AUPRC).

### 3.1.2 IMPLEMENTATION DETAILS

All models are implemented in PyTorch Paszke et al. (2019) using PyTorch Geometric Fey & Lenssen (2019). We use 3 graph set convolutional layers (`GCNConv`) with a hidden dimension of 64. For the Set Transformers model, the multi-head attention heads were set to 4, and the number of inducing points was set to 32. The DeepSets model does not use any attention mechanism. For the Set Graph Transformer model, the number of multi-head attention layers was also set to 4. A base dropout with probability 0.1 is applied after each layer to prevent overfitting, with adjustments based on hyperparameter optimisation on the BACE data set (Srivastava et al., 2014). We train using the AdamW optimiser with a learning rate of $1 \times 10^{-3}$ for DeepSets and Set Transformer and $1 \times 10^{-4}$ for GST (lowering the learning rate for DeepSets and Set Transformer was not beneficial to their performance) (Loshchilov & Hutter, 2019). All results are gathered using early stopping on the validation metric.

For CIFAR-10 and the full DDI data set, a dropout of 0.2 was used, and all models were trained for 200 epochs with a batch size of 16. For $DDI_{0.01}$, we increased the number of epochs to 2,000 and the batch size to 32.

## 4 RESULTS AND DISCUSSION

Our experiments test whether performing node-level propagation and cross-graph contextual modelling simultaneously, rather than sequentially, improves predictive performance. The evaluation results for GST, DeepSets, and Set Transformer on the CIFAR-10 data set replicate the findings of Chinello & Boracchi (2025) on non-graph data for graph data, with macro-averaged ROC-AUC increasing with set size (Table 1). In addition, we show that when trained on sets with cardinalities 5, 10, and 5-20, GST performs better than DeepSets and Set Transformer during inference on singlets. However, this training scheme does not improve the macro-averaged ROC-AUC compared to

training and inference on singlets. This suggests that the architecture fails to generalise from higher cardinalities to singlets.

Table 1: Classification macro-averaged ROC-AUC (%) on CIFAR-10 for different set sizes. {1} denotes inference on singlets.

| Data Set | $|S|$ | GCN + Deep Sets | GCN + SetTransformer | GST (ours) |
|---|---|---|---|---|
| | 10 | 98.2 ± 0.1 | 95.8 ± 0.5 | **99.1 ± 0.1** |
| CIFAR10 | 5 | 95.8 ± 0.1 | 92.5 ± 0.2 | **96.9 ± 0.0** |
| | 1 | 80.9 ± 0.2 | 80.0 ± 0.6 | **83.7 ± 0.1** |
| | 5-20 {1} | 78.9 ± 0.1 | 75.4 ± 1.3 | **81.8 ± 0.3** |
| | 10 {1} | 77.8 ± 0.3 | 66.3 ± 1.4 | **80.8 ± 0.4** |
| | 5 {1} | 79.6 ± 0.1 | 72.2 ± 0.6 | **81.6 ± 0.1** |

Evaluating GST against DeepSets and Set Transformer on molecular binary classification tasks, we show that our approach consistently performs better on set cardinalities 10 and 20. For the smaller set cardinality of 5, the gap between the methods is narrower. In addition, the standard deviations for smaller set sizes (5 and 1-5) substantially increase across the 5 runs, indicating a sensitivity of the models to set composition. However, as the sets are randomly constructed, this outcome is to be expected given the underlying structural variability of the molecules, whereby certain combinations may prove more advantageous than others. As the set size increases, this effect is likely progressively averaged out. Interestingly, when training and testing on random cardinalities between 1-5 and 1-10, the advantage of GST compared to DeepSets decreases. This again suggests that GST may be more sensitive when input cardinalities differ. Compared with DeepSets and GST, Set Transformer achieves considerably lower performance, with the gap most pronounced for sets of cardinality 20. In summary, GST performs best for larger, homogeneous set cardinalities and provides more reliable results due to a lower standard deviation. However, the advantage diminishes as the set cardinality decreases.

Table 2: ROC-AUC (%) on molecular classification benchmarks for different set sizes.

| Data Set | $|S|$ | GCN + Deep Sets | GCN + SetTransformer | GST (ours) |
|---|---|---|---|---|
| | 20 | 95.4 ± 3.0 | 70.8 ± 8.5 | **99.0 ± 0.9** |
| | 10 | 89.3 ± 5.0 | 86.6 ± 3.7 | **98.5 ± 1.2** |
| BACE | 5 | 78.0 ± 16.2 | 77.5 ± 11.4 | **87.3 ± 10.7** |
| | 1-10 | 85.6 ± 4.9 | 83.3 ± 4.9 | **87.1 ± 2.4** |
| | 1-5 | 75.8 ± 13.0 | 70.5 ± 9.8 | **78.6 ± 9.2** |
| | 20 | 97.7 ± 3.2 | 85.4 ± 15.8 | **99.5 ± 0.8** |
| | 10 | 94.5 ± 6.3 | 92.0 ± 6.3 | **99.2 ± 1.1** |
| BBBP | 5 | 82.9 ± 16.6 | 80.7 ± 18.2 | **84.4 ± 17.8** |
| | 1-10 | **91.8 ± 7.5** | 88.0 ± 6.1 | 91.2 ± 6.0 |
| | 1-5 | 77.1 ± 15.9 | 75.2 ± 14.7 | **78.1 ± 16.4** |
| | 20 | 98.8 ± 2.5 | 92.7 ± 13.3 | **99.6 ± 0.6** |
| | 10 | 97.1 ± 5.2 | 95.9 ± 6.0 | **99.5 ± 0.9** |
| Pgp Broccatelli | 5 | 87.4 ± 15.8 | 85.8 ± 17.4 | **88.5 ± 16.4** |
| | 1-10 | **93.6 ± 6.1** | 90.0 ± 5.2 | 92.3 ± 5.2 |
| | 1-5 | 82.2 ± 15.8 | 80.3 ± 15.0 | **82.4 ± 15.5** |
| | 20 | 98.5 ± 2.8 | 90.3 ± 14.6 | **99.5 ± 0.7** |
| | 10 | 96.2 ± 5.7 | 94.6 ± 6.3 | **99.4 ± 0.9** |
| BBB Martins | 5 | 85.4 ± 16.2 | 83.6 ± 17.9 | **86.6 ± 17.0** |
| | 1-10 | **92.4 ± 6.6** | 89.1 ± 5.5 | 90.8 ± 5.0 |
| | 1-5 | 79.6 ± 15.7 | 77.6 ± 14.5 | **80.4 ± 15.9** |
| | 20 | - | - | - |
| | 10 | - | - | - |
| CYP3A4 | 5 | 62.6 ± 6.1 | 68.6 ± 6.6 | **76.8 ± 3.0** |
| | 1-10 | - | - | - |
| | 1-5 | 63.2 ± 2.2 | 62.0 ± 5.4 | **70.2 ± 2.9** |

Finally, we consider the drug-drug interaction (DDI) data set. GST and Set Transformer achieve comparable performance, whereas DeepSets underperforms (Table 3). However, while the accuracy and F1-score are almost identical between GST and Set Transformer, the gap widens by a factor of

10 for AUPRC. This suggests that GST may provide superior ranking of interaction types between drugs, which is important given an imbalance between the interaction types (Ryu et al., 2018). To investigate further, we randomly subsample the training, validation, and test sets by a factor of 0.01 to evaluate the behaviour of the models under data sparsity. Under the undersampled data regime, the performance gap between Set Transformer and GST substantially increases, whereas the gap to Deep Sets decreases. We interpret this result as evidence that GST is more data-efficient than the alternatives, potentially due to its simultaneous feature extraction and contextualisation, which enables the extraction of more information from limited data. This observation is consistent with the observations in Table 2, where the performance gains of GST increase in the smaller data set CYP3A4.

Table 3: Performance on DDI prediction benchmarks with different evaluation metrics ($|S| = 2$).

| Data Set | Metric | $|S|$ | GCN + Deep Sets | GCN + SetTransformer | GST (ours) |
|---|---|---|---|---|---|
| DDI$_{Full}$ | Accuracy | 2 | 81.3 ± 0.2 | 94.3 ± 0.1 | **94.5 ± 0.1** |
| | F1 | 2 | 67.1 ± 0.9 | 87.5 ± 0.6 | **87.7 ± 0.5** |
| | AUPRC | 2 | 73.6 ± 0.5 | 86.9 ± 0.5 | **88.9 ± 0.2** |
| DDI$_{0.01}$ | Accuracy | 2 | 48.3 ± 1.0 | 44.2 ± 1.0 | **54.2 ± 0.6** |
| | F1 | 2 | 18.3 ± 1.4 | 15.5 ± 0.4 | **21.5 ± 1.3** |
| | AUPRC | 2 | 37.5 ± 1.8 | 35.9 ± 0.8 | **41.3 ± 0.9** |

We interpret this result as a preliminary indication that the core assumption behind GST, namely that combining feature extraction and set contextualisation, is not only beneficial when applied to image data or synthetic sets, as shown in Tables 1 and 2, respectively, but also on real-world molecular data.

## 5 CONCLUSION AND FURTHER WORK

We introduced the graph set transformer (GST), an architecture that simultaneously performs node-level feature propagation and set-level contextual modelling, rather than cascading a graph encoder with a set aggregator such as DeepSets or Set Transformer. Across a series of benchmarks, our approach consistently outperforms the alternatives, with increasing performance advantages as set cardinality grows. In addition, GST demonstrates strengths in low- and sparse-data regimes, potentially highlighting the advantages of simultaneous feature extraction and contextualisation. However, several limitations warrant discussion. First, molecular classification benchmarks on randomly constructed sets, rather than naturally occurring ones, may fail to highlight the advantages of the architecture due to limited intra-set context. Furthermore, the fact that GST struggles to generalise from training on larger set cardinalities to inference on singlets suggests that adjustments to the architecture are required to allow potentially valuable set context to boost singlet prediction performance.

Future work should prioritise the evaluation of the architecture on naturally occurring molecular sets. Specifically, conformer ensembles, where the incorporation into molecular geometry-aware graph convolutional layers is required. Furthermore, as the architecture is broadly applicable to data in which sets of graphs naturally arise, extensions to protein conformation ensembles, chemical or biochemical reactions, or metabolic pathways represent promising directions.

### MEANINGFULNESS STATEMENT

The graph set transformer introduced in this work enables more effective modelling of molecular sets by capturing within-set context between related compounds. Beyond synthetic benchmarks and drug-drug interactions, GST has potential applications wherever sets of biological entities representable as graphs occur, such as conformer ensembles of small molecules or even proteins. We consider this architecture a meaningful addition to the computational toolset for the representation of life, as it can be applied across various contexts in chemistry and biology.

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
