# OpenReview forum: "Graph Set Transformer: Learning Graph Representations with Set Context"
_ICLR.cc/2026/Workshop/LMRL — ICLR 2026 Workshop LMRL Poster_

### Official Review · Reviewer_ApKy · 2026-02-15
**Graph Set Transformer: Promising Set-Context Graph Learning**

**Rating:** 6
**Confidence:** 4

**Review:**

The paper proposes the Graph Set Transformer (GST), which learns over sets of graphs by jointly performing node-level message passing and set-level contextualization within each layer. The design is intuitive, and the reported improvements suggest potential gains in data efficiency over two-stage baselines.

# Strength

1. The paper presents a clear architecture: integrate set-level attention at intermediate GNN layers and inject it back into node features through a gating mechanism.

2. Empirical gains suggest potential for better data efficiency.

# Weaknesses

1. For evaluation on molecular benchmarks, the paper constructs sets by randomly grouping molecules from datasets that do not naturally define graph sets. This weakens the central motivation that graphs within a set share higher-level semantics; experiments on tasks with naturally occurring graph sets (e.g., conformer ensembles, protein conformation ensembles, reaction/pathway sets) would better demonstrate practical relevance.

2. No ablation studies isolate the contributions of attention vs. pooling vs. gating, test the broadcasting strategy (injecting set context at all layers vs. only the last), or assess sensitivity to the graph pooling operator (max vs. sum/mean, commonly used in graph classification).

3. Several formulas and definitions are incomplete. The equation around Line 150 appears to have a typo.

---

### Official Review · Reviewer_LCQT · 2026-02-20

**Rating:** 5
**Confidence:** 3

**Review:**

This paper introduces the Graph Set Transformer, a neural architecture designed to learn representations from sets of graphs by jointly modeling intra-graph structure and inter-graph context. Unlike prior approaches that rely on precomputed graph embeddings followed by set-level aggregation, the proposed method integrates node-level message passing with cross-graph attention in a single end-to-end framework. A gating mechanism is used to fuse local graph features with set-level contextual information during representation learning. The model is evaluated on multiple molecular classification benchmarks, where it achieves consistently strong ROC-AUC performance across varying set cardinalities. Additional experiments on a drug-drug interaction task demonstrate notable improvements in F1 score under sparse data conditions. Overall, the work argues that contextualizing graphs within sets can substantially improve performance when graphs share higher-level semantic structure.

**Strengths**

1. Proposes a clear architectural contribution that removes the bottleneck between graph embedding and set-level contextualization present in prior methods.
2. Demonstrates strong empirical performance across several benchmarks and set sizes, including gains in challenging low-data regimes.
3. Presents a conceptually simple and general framework that can be combined with standard graph convolution operators.

**Weaknesses**

1. The architecture builds on standard GCN layers and attention mechanisms, and the novelty primarily lies in their integration rather than in new modeling primitives.
2. Experimental evaluation is limited to molecular and drug interaction domains, leaving generalization to other graph-set tasks untested.
3. Computational complexity and scalability with respect to large graphs or large set cardinalities are not analyzed in detail.
4. The choice of global max pooling is fixed, and the impact of alternative pooling strategies is not explored.
5. Ablation studies isolating the contribution of the gating mechanism and cross-graph attention are limited or not clearly presented.
6. Alignment to LMRL topics is low.

---

### Official Review · Reviewer_YLDn · 2026-02-26
**Graph Set Transformer**

**Rating:** 6
**Confidence:** 3

**Review:**

This paper introduces the Graph Set Transformer (GST), an architecture that simultaneously performs node-level feature propagation and cross-graph set contextualization, eliminating the bottleneck of separate graph encoding and set aggregation in existing methods (DeepSets/Set Transformer). GST is validated on CIFAR-10 (graph-encoded) and molecular classification/DDI benchmarks, achieving superior performance on large set cardinalities and sparse data. The work is innovative and addresses a real gap in set-level graph learning, though it has limitations in natural set design and generalization to singlet inference.

Pros
End-to-end GST architecture that simultaneously performs graph convolution and set attention, eliminating the bottleneck of separate graph encoding and set aggregation.
Node-specific gating mechanism that effectively blends local node features and global set context, adapted to the structural heterogeneity of graphs.
Comprehensive validation on both graph-encoded image data and molecular benchmarks, showing consistent performance gains on large set cardinalities.
﻿
Cons
The molecular sets used in experiments are randomly constructed rather than naturally occurring (e.g., conformer ensembles, scaffold-based molecule sets), which limits the practical relevance of the results – the benefit of set context is less pronounced in random sets.
GST struggles to generalize from training on large set cardinalities to inference on singlets, a critical limitation for real-world applications where singlet prediction is common.

---

### Meta-Review · Area_Chair_aqoA · 2026-02-27

**Recommendation:** Accept (Poster)
**Confidence:** 4

**Metareview:**

Accept.

---

### Decision · Program_Chairs · 2026-03-02

**Decision:**

Accept (Poster)

**Comment:**

Please see the meta-review.